# Gut-Microbiota-Derived Metabolites and Probiotic Strategies in Colorectal Cancer: Implications for Disease Modulation and Precision Therapy

**DOI:** 10.3390/nu17152501

**Published:** 2025-07-30

**Authors:** Yi-Chu Yang, Shih-Chang Chang, Chih-Sheng Hung, Ming-Hung Shen, Ching-Long Lai, Chi-Jung Huang

**Affiliations:** 1Department of Family and Community Medicine, Cathay General Hospital, Taipei City 106438, Taiwan; yangy9999@hotmail.com; 2Division of Colorectal Surgery, Department of Surgery, Cathay General Hospital, Taipei City 106438, Taiwan; cgh06719@cgh.org.tw; 3School of Medicine, College of Medicine, Fu Jen Catholic University, New Taipei 242062, Taiwan; 4Division of Gastroenterology, Department of Internal Medicine, Cathay General Hospital, Taipei 106438, Taiwan; 5Department of Surgery, Fu Jen Catholic University Hospital, New Taipei City 243089, Taiwan; 6Division of Basic Medical Sciences, Department of Nursing, Chang Gung University of Science and Technology, Taoyuan City 333324, Taiwan; 7Center for Drug Research and Development, Chang Gung University of Science and Technology, Taoyuan City 333324, Taiwan; 8Department of Medical Research, Cathay General Hospital, Taipei City 106438, Taiwan; 9Department of Biochemistry, National Defense Medical Center, Taipei City 114201, Taiwan

**Keywords:** gut microbiota, short-chain fatty acids (SCFAs), probiotics, colorectal cancer (CRC), microbiome therapeutics

## Abstract

The human gut microbiota significantly influences host health through its metabolic products and interaction with immune, neural, and metabolic systems. Among these, short-chain fatty acids (SCFAs), especially butyrate, play key roles in maintaining gut barrier integrity, modulating inflammation, and supporting metabolic regulation. Dysbiosis is increasingly linked to diverse conditions such as gastrointestinal, metabolic, and neuropsychiatric disorders, cardiovascular diseases, and colorectal cancer (CRC). Probiotics offer therapeutic potential by restoring microbial balance, enhancing epithelial defenses, and modulating immune responses. This review highlights the physiological functions of gut microbiota and SCFAs, with a particular focus on butyrate’s anti-inflammatory and anti-cancer effects in CRC. It also examines emerging microbial therapies like probiotics, synbiotics, postbiotics, and engineered microbes. Emphasis is placed on the need for precision microbiome medicine, tailored to individual host–microbiome interactions and metabolomic profiles. These insights underscore the promising role of gut microbiota modulation in advancing preventive and personalized healthcare.

## 1. Introduction

The human gastrointestinal tract harbors a complex microbial ecosystem, collectively known as the gut microbiota, which plays a vital role in maintaining host health. In recent years, increasing attention has been given to its role in colorectal cancer (CRC), one of the most prevalent malignancies worldwide. [1,2,3]. Among these, the bacterial community dominates in number and function and particularly members of the *Firmicutes*, *Bacteroidetes*, *Actinobacteria*, and *Proteobacteria phyla* [4]. These microbial communities are essential for maintaining host health through their involvement in nutrient metabolism, vitamin synthesis (e.g., vitamin K and B12), immune system development, and protection against pathogenic organisms [4,5,6].

Recent advancements in high-throughput sequencing and metagenomic profiling have dramatically expanded our understanding of the gut microbiota, transitioning from culture-based studies to a comprehensive molecular characterization of microbial communities [7,8,9]. These technologies have unveiled not only the taxonomic diversity but also the functional capabilities of the gut microbiota, leading to the identification of core microbiomes and disease-specific microbial signatures [10]. In other words, research has increasingly recognized the gut microbiota as a key modulator of host physiology, with far-reaching effects on the gastrointestinal, immune, metabolic, and nervous systems [11,12,13,14,15]. For example, Qin et al. (2010) and Chanda et al. (2024) provided foundational metagenomic insights into gut microbial gene richness and its association with obesity [16,17].

Among their diverse functional outputs, microbial metabolites, and particularly short-chain fatty acids (SCFAs—acetate, propionate, and butyrate, for example), have emerged as critical mediators of host physiology (Table 1) [18]. SCFAs are generated through the fermentation of dietary fibers by commensal bacteria and are vital for maintaining gut barrier integrity, regulating immune responses, and modulating energy homeostasis [19,20,21]. Notably, reduced levels of SCFAs, particularly butyrate, have been associated with colorectal tumorigenesis and impaired mucosal immunity. Beyond their local actions within the gastrointestinal tract, SCFAs exert systemic effects by regulating gene expression through epigenetic mechanisms, most notably via the inhibition of histone deacetylases (HDACs) [22,23]. Their well-documented antioxidant and anti-inflammatory properties underscore their potential as therapeutic agents across a spectrum of inflammatory and metabolic disorders [21,23,24].

In addition to SCFAs, the gut microbiota produces a broad repertoire of metabolites, including bile acids, tryptophan derivatives, neurotransmitters (e.g., serotonin and gamma-aminobutyric acid [GABA]), and trimethylamine-N-oxide (TMAO) [25,26,27,28]. These bioactive compounds collectively shape host metabolic and immune landscapes. For example, the gut–brain axis has emerged as a well-established paradigm describing how microbial signaling influences central nervous system function and behavior via neuroimmune and neuroendocrine pathways [29].

This rapidly expanding body of knowledge has spurred the development of novel microbiota-targeted therapies, including probiotics, prebiotics, synbiotics, postbiotics, and fecal microbiota transplantation [30,31]. These interventions aim to restore or modulate microbial composition and function to promote health and treat disease [32]. In parallel, the field of precision microbiome medicine has gained momentum, emphasizing individualized therapeutic strategies tailored to the host’s unique microbial and metabolic profiles [33].

In this narrative review, we synthesize current evidence on the physiological roles of the gut microbiota and its metabolites, with a particular emphasis on SCFAs, especially butyrate. Given its strong relevance to CRC, we highlight CRC as a central example of how microbial metabolites and probiotics influence host health and disease progression. While other organ systems are discussed to illustrate broader implications, CRC serves as a mechanistically grounded disease model. The narrative format allows us to integrate emerging preclinical and clinical findings with evolving therapeutic concepts.

**Table 1 nutrients-17-02501-t001:** Functions of major SCFAs and their host physiological effects.

SCFA ^1^	Primary Function	Physiological Effects ^2^	Reference
Acetate	Substrate for lipogenesis and cholesterol synthesis	Regulates appetite via hypothalamic signaling, contributes to anti-inflammatory effects via GPR43 activation	[34,35]
Propionate	Gluconeogenic substrate in the liver	Lowers blood glucose and inhibits cholesterol synthesis via PPAR-γ activation; modulates intestinal gluconeogenesis	[36,37]
Butyrate	Primary energy source for colonocytes	Promotes Treg differentiation via HDAC inhibition; enhances epithelial barrier integrity; anti-cancer via gene regulation	[18,19,38,39]

^1^ SCFA, short-chain fatty acid. ^2^ GPR, G protein-coupled receptor; HDAC, histone deacetylase; PPAR, peroxisome proliferator-activated receptor; Treg, regulatory T cell.

Microbial dysbiosis, defined as a disturbance in the composition or function of the gut microbiota, has been increasingly linked to a wide range of diseases, including gastrointestinal disorders, metabolic syndromes, neuropsychiatric conditions, cardiovascular diseases, and notably CRC (Table 2) [40,41,42]. Mounting evidence supports a protective role for SCFAs, particularly butyrate, in CRC suppression [41,43,44,45].

## 2. Methodology and Analysis

This narrative review synthesizes the current scientific knowledge on the physiological functions of the gut microbiota, microbial-derived SCFAs, and probiotic therapies across multiple disease contexts. The review emphasizes mechanistic insights, clinical applications, and future therapeutic directions.

### 2.1. Literature Search Strategy

This manuscript is designed as a narrative review, not a systematic review. A structured literature search was conducted using PubMed, Scopus, and Web of Science, covering studies published between January 2010 and May 2025. Search terms included “*gut microbiota*”, “*short-chain fatty acid*”, “*butyrate*”, “*probiotics*”, “*postbiotics*”, “*synbiotics*”, “*colorectal cancer*”, “*gut-brain axis*”, “*immune modulation*”, and “*microbiome therapeutics*”. Boolean operators and MeSH terms were used to refine the results. Only English-language publications were considered.

### 2.2. Literature Overview Approach

Eligible studies included peer-reviewed original research articles, meta-analyses, and narrative reviews that addressed topics related to microbial metabolites, particularly SCFAs, as well as gut–host interactions and probiotic-based interventions. Studies were considered if they involved human participants, animal models, or in vitro experiments with clear relevance to host physiological or pathological outcomes.

Studies were excluded if they focused exclusively on non-bacterial components of the microbiota (such as fungi or viruses) without addressing bacterial-derived metabolites or gut–host crosstalk. In addition, publications that lacked sufficient mechanistic detail or clinical relevance to SCFA metabolism, microbial dysbiosis, or probiotic functionality were omitted from the review.

### 2.3. Analytical Approach

An initial screening of database results yielded approximately 900 citations. After removing duplicates and excluding studies based on title and abstract relevance, a total of 210 articles were selected for full-text assessment. From these, approximately 160 core articles were retained for in-depth review and synthesis. The selected studies were then thematically categorized and critically evaluated to ensure relevance and scientific rigor.

The thematic analysis centered on several key domains: distinctions between homeostatic and dysbiotic microbiome states; the mechanistic roles of SCFAs and their clinical implications; organ-specific effects of probiotics across gastrointestinal, metabolic, neuropsychiatric, and renal systems; butyrate’s anti-tumor activity in the context of CRC; and the current limitations and emerging potential of microbial-based therapeutics. This structured approach enabled an integrated synthesis of evidence from both the experimental and clinical literature, facilitating a comprehensive and translational overview of microbiota-related health interventions.

As a narrative review, this work inherently carries a risk of selection bias. To mitigate this, studies were selected based on scientific quality, relevance to mechanistic pathways, and clinical impact. CRC was chosen as the primary focus for deeper exploration due to its well-established microbiota–metabolite links, while additional organ systems were included to provide broader context.

## 3. Gut Microbiota in Health and Disease

### 3.1. Homeostatic Functions in the Healthy Host

In a healthy individual, the gut microbiota plays a central role in maintaining host homeostasis by supporting a range of physiological processes [50]. One of its key functions is the fermentation of dietary fibers into SCFAs, such as acetate, propionate, and butyrate. These SCFAs serve not only as vital energy sources for colonocytes but also as signaling molecules involved in regulating immune responses and metabolic functions [18].

The microbiota also contributes to micronutrient biosynthesis, producing essential vitamins such as vitamin K and certain B vitamins, including B12 [51,52]. Another critical function is the reinforcement of the intestinal barrier. Commensal bacteria stimulate the expression of tight junction proteins, such as claudin-1 and occludin, thereby enhancing epithelial integrity and preventing microbial translocation [21,53].

Importantly, the gut microbiota orchestrates immune balance by modulating both pro- and anti-inflammatory pathways, ensuring effective defense without excessive immune activation [43]. Under balanced microbial conditions, these regulatory mechanisms help sustain mucosal immunity, support metabolic homeostasis, and defend against pathogenic invasion [54].

Notably, specific commensal taxa, such as *Faecalibacterium prausnitzii*, *Bifidobacterium adolescentis*, and *Roseburia* spp., are recognized for their anti-inflammatory properties [55,56]. These microbes produce SCFAs and other metabolites that sustain an anti-inflammatory tone, reinforce epithelial function, and promote regulatory immune responses, further solidifying the microbiota’s role in host health [57,58].

### 3.2. Microbiota–Metabolite Interactions Across Host Systems

The disruption of microbial homeostasis (also termed dysbiosis) has been increasingly implicated in the pathogenesis of a wide range of diseases [59].

#### 3.2.1. Gastrointestinal and Metabolic Disorders

In the gastrointestinal tract, dysbiosis is closely associated with inflammatory bowel disease (IBD), celiac disease, and CRC [60,61,62]. These conditions are often characterized by a bloom of pro-inflammatory taxa such as *Escherichia coli* and a depletion of beneficial butyrate-producing microbes like *Faecalibacterium prausnitzii* [55,63]. Such alterations compromise epithelial barrier integrity and trigger mucosal immune activation, leading to chronic intestinal inflammation [40,48,64,65].

Beyond the gut, metabolic disorders such as obesity, type 2 diabetes, and hypertension have been associated with reduced microbial diversity and diminished SCFA production [66,67]. Dysbiosis contributes to low-grade systemic inflammation, insulin resistance, and metabolic dysregulation [47,66,68,69,70]. SCFA deficiency has been shown to downregulate epithelial tight junction proteins, increasing intestinal permeability and facilitating microbial translocation [71]. This, in turn, activates pro-inflammatory signaling pathways, such as TLR4–NFκB, perpetuating chronic inflammation [72].

#### 3.2.2. Neuroimmune and CNS Disorders

The gut–brain axis offers a compelling framework to understand the neuropsychiatric impacts of dysbiosis. Microbes contribute to the synthesis of neuroactive compounds, including GABA, serotonin, and tryptophan-derived metabolites, which influence mood, cognition, and behavior [73,74,75]. Disruption of this axis, through a reduced abundance of neurotransmitter-producing bacteria, has been associated with depression, anxiety, autism spectrum disorders, and neurodegenerative diseases such as Alzheimer’s and Parkinson’s [76,77,78].

#### 3.2.3. Cardiovascular and Renal Systems

Cardiovascular disease represents another systemic outcome of dysbiosis. Increased intestinal permeability and SCFA deficiency promote microbial translocation and systemic inflammation, which elevate circulating levels of pro-atherogenic metabolites such as TMAO. This contributes to endothelial dysfunction and vascular inflammation [79,80]. Additionally, similar mechanisms, such as molecular mimicry and aberrant immune stimulation, may drive autoimmune diseases in genetically predisposed individuals [81].

In summary, microbial dysbiosis is a critical and multifaceted contributor to a spectrum of systemic diseases, mediated through impaired barrier function, altered metabolite production, chronic inflammation, and immune dysregulation.

### 3.3. The Pivotal Role of SCFAs

SCFAs, primarily acetate, propionate, and butyrate, act as essential molecular intermediaries between the gut microbiota and host physiology. Each SCFA exhibits distinct bioactivity. Acetate contributes to appetite regulation and lipid metabolism via GPR43-mediated signaling in the hypothalamus and adipose tissue, thereby influencing energy balance and immune responses [82,83]. Propionate serves as a gluconeogenic substrate in the liver and modulates both cholesterol synthesis and glycemic control, partly through PPAR-γ activation [84]. Among these, butyrate has been the most extensively studied due to its potent immunomodulatory and anti-cancer properties. It promotes the differentiation of regulatory T cells (Treg) through HDAC inhibition, which enhances Foxp3 expression, a key transcription factor for Treg development [85,86]. Butyrate also reinforces epithelial barrier integrity, suppresses pro-inflammatory signaling, and downregulates oncogenic targets such as CSE1L and PLAC8, contributing to its protective effects in CRC [38,41].

Through these mechanisms, SCFAs play a central role in regulating metabolism, immune homeostasis, and barrier function. In addition to local intestinal effects, SCFAs exert systemic actions by influencing gene expression, signaling pathways, and immune responses across distant organs [21]. In the immune system, butyrate promotes the differentiation of Treg cells by inhibiting histone deacetylases (HDACs), leading to upregulated expression of Foxp3, a key transcription factor for Treg development [22,86]. Additionally, SCFAs activate GPCRs such as GPR43 and GPR109A, which are expressed on epithelial cells, dendritic cells, and macrophages, contributing to anti-inflammatory cytokine production (e.g., IL-10) and barrier protection [87,88]. Acetate and propionate also interact with GPR41, influencing gut motility and sympathetic nervous activity [87,89]. Butyrate is transported into colonocytes primarily via the SLC5A8 transporter, facilitating its intracellular HDAC inhibitory activity.

The physiological impact of SCFA deficiency has been linked to multiple disease processes. In the urinary system, a decline in SCFA-producing microbes contributes to increased gut permeability, systemic inflammation, and subsequent renal and bladder dysfunction [90,91]. In the context of neurodegenerative diseases such as Parkinson’s disease (PD), altered SCFA levels are associated with neuroinflammation and the aggregation of α-synuclein, providing a mechanistic connection between gut dysbiosis and central nervous system pathology [21]. In cardiovascular health, SCFAs support vascular homeostasis and lipid regulation, and their depletion correlates with heightened cardiometabolic risk [80]. Furthermore, the loss of SCFA-producing species weakens epithelial junctions and elevates circulating lipopolysaccharide levels, fostering metabolic endotoxemia and chronic inflammation [92].

SCFA–receptor interactions are not only tissue-specific but also dose-dependent, underscoring the importance of metabolite concentration gradients [18]. Moreover, the butyrate–SLC5A8–HDAC axis and SCFA–GPCR signaling represent complementary mechanisms that converge on immune modulation, barrier integrity, and metabolic regulation [57,93]. These receptor-mediated and epigenetic mechanisms provide the molecular basis for the therapeutic potential of SCFAs described below.

Collectively, these findings underscore the multifaceted roles of SCFAs in maintaining physiological equilibrium across multiple organ systems. Their centrality in inflammation control, metabolic regulation, immune signaling, and barrier maintenance highlights their potential as promising therapeutic targets across diverse disease contexts, especially CRC (see Table 3).

## 4. Probiotics and Therapeutic Microbial Modulation

### 4.1. Core Mechanisms of Probiotic Action

Probiotics are live microorganisms that confer health benefits to the host through a variety of synergistic mechanisms. A fundamental function of probiotics is the competitive exclusion of pathogens. By occupying mucosal adhesion sites and consuming available nutrients, probiotics inhibit the colonization and overgrowth of harmful microbes, thereby mitigating infection and inflammation [94,95,96,97].

Probiotics also play a critical role in reinforcing intestinal barrier integrity. They upregulate the expression of tight junction proteins such as occludin and claudins, enhance mucin secretion, and reduce intestinal permeability, thereby limiting microbial translocation and systemic inflammatory responses [20,46,75].

In addition, immune modulation is a hallmark of probiotic function. Probiotics stimulate dendritic cells and promote the differentiation of Treg cells [98,99]. These actions increase the production of anti-inflammatory cytokines, such as IL-10 and TGF-β, and help maintain immune tolerance, which is particularly beneficial in chronic inflammatory and autoimmune diseases [18,21,81].

SCFAs generated by probiotic-induced butyrogenic bacteria further support immune tolerance through HDAC inhibition, which epigenetically enhances Foxp3 expression and Treg cell differentiation [85,86]. SCFAs also activate GPCRs (GPR43, GPR41) on immune and epithelial cells, reinforcing the anti-inflammatory signaling cascade [87].

Separately, probiotics support metabolic health by promoting the growth of butyrate-producing commensals such as *Faecalibacterium* and *Roseburia*. This leads to an enhanced production of SCFAs, especially butyrate, which contribute to mucosal homeostasis, immune regulation, and systemic metabolic balance [100].

Certain well-studied strains exhibit targeted benefits. For example, *Lactobacillus rhamnosus* GG has been shown to reduce inflammation in IBD [101], while *Bifidobacterium longum* improves gut–brain axis signaling through vagal nerve modulation and cytokine regulation, ultimately alleviating symptoms of anxiety and improving gastrointestinal function [102,103,104]. These strains also facilitate SCFA production by encouraging the proliferation of butyrogenic microbiota [105,106].

### 4.2. Probiotic Interventions in Disease-Specific Contexts

Probiotics have demonstrated broad clinical utility across multiple organ systems, supported by a growing body of evidence from both randomized controlled trials and mechanistic studies. Having outlined the general mechanisms through which probiotics and microbial metabolites, particularly SCFAs, confer health benefits, this section now examines their roles in specific disease contexts [18]. To enhance clarity and depth, representative conditions from distinct physiological systems are grouped and discussed individually. In particular, CRC, inflammatory bowel disease (IBD), and PD are highlighted for their well-established links to microbial dysbiosis and SCFA-mediated regulation [15,19,21,43,107].

#### 4.2.1. CRC

Probiotic interventions in CRC have shown promising outcomes through the modulation of gut microbial composition and the restoration of SCFA production, particularly butyrate [48]. Preclinical studies demonstrate that butyrate-producing strains, such as *Clostridium butyricum* and *Butyricicoccus pullicaecorum*, reduce tumor burden by reinforcing epithelial barrier integrity, suppressing pro-inflammatory cytokines, and inducing apoptosis in colonic epithelial cells [41,108]. Butyrate’s effects are mediated via HDAC inhibition, which upregulates Foxp3 expression and supports Treg cell development, a critical factor in immune homeostasis [86]. Additionally, butyrate has been shown to suppress oncogenic pathways, including PLAC8 and CSE1L, which are implicated in CRC progression [38,41].

Clinical investigations have explored multi-strain probiotic formulations (e.g., *Lactobacillus rhamnosus* GG, *Bifidobacterium longum*) as adjuncts in CRC patients undergoing surgery or chemotherapy [109,110]. Reported benefits include reduced postoperative infection rates, improved mucosal cytokine profiles, and enhanced patient recovery [111]. However, human studies remain heterogeneous in design, with variation in probiotic strains, dosage, and outcome measures, highlighting the need for standardized CRC-focused trials.

#### 4.2.2. Probiotic Applications in Inflammatory and Functional Bowel Disorders

In gastrointestinal disorders such as inflammatory bowel disease (IBD) [94,112], irritable bowel syndrome (IBS) [113,114], and antibiotic-associated diarrhea, probiotics have shown efficacy in alleviating symptoms, promoting mucosal healing, and sustaining clinical remission [46,75]. For instance, trials using *Lactobacillus plantarum* and *Bifidobacterium breve* in IBS demonstrated significant reductions in bloating and abdominal discomfort [115].

In the context of IBD specifically, probiotics play a critical role in restoring microbial diversity, reinforcing intestinal barrier function, and modulating chronic mucosal inflammation [58]. Several probiotic formulations, such as the multi-strain mixture VSL#3, have shown beneficial effects in both ulcerative colitis (UC) and Crohn’s disease (CD) [116]. These probiotics enhance butyrate production, which in turn strengthens epithelial tight junctions and reduces intestinal permeability [117]. Mechanistically, butyrate suppresses NF-κB signaling and promotes Treg differentiation through HDAC inhibition, leading to improved immune tolerance [93]. Probiotic-induced microbial shifts are associated with reduced pro-inflammatory cytokines (e.g., IL-6, TNF-α, IL-17) and increased levels of anti-inflammatory mediators such as IL-10 and TGF-β [34,73]. Although results from clinical trials vary, probiotic therapy has shown more consistent efficacy in maintaining remission in UC than in CD [118].

#### 4.2.3. PD

Emerging evidence links gut dysbiosis and reduced SCFA levels with the progression of PD via the gut–brain axis [21]. In PD, probiotic interventions aim to correct microbial imbalance, reduce intestinal inflammation, and potentially modulate neuroinflammation and motor symptoms [119].

Preclinical studies indicate that probiotic supplementation can restore SCFA-producing taxa (e.g., *Faecalibacterium prausnitzii*, *Roseburia*), which are often depleted in PD patients [120]. The resulting increase in butyrate levels contributes to improved epithelial barrier integrity and reduced systemic endotoxemia, which may lower the peripheral inflammatory signals implicated in α-synuclein aggregation in the brain [121]. Furthermore, SCFAs may act on microglial cells and enteric neurons through GPR43 and GPR109A signaling, promoting neuroprotection [21].

Although human clinical data remain limited, several small-scale trials report improvements in gastrointestinal function, constipation symptoms, and inflammatory markers following probiotic administration in PD patients, suggesting potential as an adjunctive therapeutic strategy [122].

#### 4.2.4. Other Systemic Disorders

In metabolic syndrome, probiotics have been shown to improve lipid profiles, lower blood pressure, and reduce low-density lipoprotein (LDL) cholesterol, while also attenuating obesity-related inflammation [47]. These effects are believed to be mediated by enhanced SCFA production and modulation of systemic immune responses.

In chronic kidney disease (CKD), probiotic supplementation has been reported to restore intestinal barrier function, reduce microbial translocation, and lower the burden of uremic toxins, thereby mitigating systemic inflammation and improving both renal and cardiovascular outcomes [100].

Neuropsychiatric disorders have also emerged as a potential target for probiotic therapy. Recent studies suggest that modulation of the gut–brain axis by probiotics may influence neurotransmitter synthesis (e.g., GABA, serotonin) and systemic cytokine levels [107,123]. Although early clinical results are promising, heterogeneity in outcomes is likely due to interindividual differences in gut microbiota composition.

Collectively, these findings underscore the systemic reach of probiotic interventions, while highlighting the importance of precision approaches tailored to individual microbiota profiles.

## 5. Butyrate and CRC

### 5.1. Microbial Metabolite–Host Signaling Pathways in Tumor Suppression

Among SCFAs, butyrate stands out for its multifaced anti-cancer properties [124]. Particularly in the context of CRC, substantial research has explored butyrate’s roles in intestinal homeostasis, mucosal health, the regulation of inflammation, and tumor suppression [125].

As a pleiotropic metabolite, butyrate modulates tumor biology through both epigenetic and receptor-mediated mechanisms, notably by inducing apoptosis and cell cycle arrest in cancer cells [126]. A key anti-tumor mechanism involves HDAC inhibition, which contributes to chromatin remodeling and gene expression regulation [124,127]. As illustrated in Figure 1, butyrate enhances the function of tumor suppressor proteins such as p53 and suppresses oncogenic drivers like CSE1L and PLAC8, thereby curtailing malignant transformation and proliferation [38,41].

Beyond direct cytotoxic effects, butyrate plays a crucial role in preserving intestinal barrier integrity, promoting epithelial cell differentiation, and facilitating mucosal repair [128,129]. Foundational studies by Donohoe et al. (2011) and Chang et al. (2020) have demonstrated that butyrate can simultaneously induce apoptosis and cell cycle arrest in CRC cells while also enhancing mucosal healing and suppressing pro-inflammatory cytokines, thereby providing protection against both local and systemic inflammation [43,130]. In addition, butyrate inhibits pro-metastatic signaling pathways, contributing to the attenuation of cancer progression [38,41,43].

Emerging evidence from preclinical models highlights the anti-cancer potential of specific butyrate-producing bacteria, such as *Clostridium butyricum* and *Butyricicoccus pullicaecorum* (Table 4). These commensals exert effects via both metabolic production and receptor-mediated signaling. Butyrate sends signals through G-protein-coupled receptors (GPCRs), particularly GPR41 and GPR43, which are essential for epithelial barrier maintenance and immune modulation [43,94,108]. The activation of these receptors influences immune homeostasis, energy regulation, and epithelial resilience [131,132]. Moreover, SCFA transporters such as SLC5A8 facilitate butyrate uptake into colonic epithelial cells, enhancing its intracellular signaling capacity [133,134]. These coordinated pathways reinforce the physiological and anti-tumor functions of butyrate.

Beyond SCFAs, microbial metabolites derived from tryptophan catabolism, including indole-3-aldehyde, indole-3-propionic acid, and indole-3-acetic acid, also play important roles in mucosal immune regulation [135]. These metabolites act primarily through activation of the aryl hydrocarbon receptor (AhR), a ligand-activated transcription factor expressed in epithelial cells and innate lymphoid cells (ILCs) [85]. Upon activation, AhR signaling induces IL-22 production, which promotes epithelial regeneration, enhances antimicrobial peptide secretion, and supports goblet cell differentiation [136]. These immunological effects are particularly relevant in chronic inflammatory diseases such as IBD and CRC [137,138]. Additionally, SCFAs may indirectly support this pathway by shaping the composition of tryptophan-metabolizing microbiota, thereby preserving mucosal immune homeostasis [139].

Collectively, these findings underscore the central role of butyrate as a tumor-suppressive metabolite and provide strong mechanistic support for therapeutic strategies aimed at restoring SCFA production through microbiota-targeted interventions (see Figure 1 and Table 4).

### 5.2. Dietary and Functional Approaches

Dietary and nutritional strategies represent a promising avenue for enhancing endogenous butyrate production and reducing the risk of CRC [19,38]. Prebiotics, such as inulin and resistant starch, as well as omega-3 polyunsaturated fatty acids, have been shown to selectively stimulate the growth of butyrate-producing bacteria, thereby strengthening epithelial integrity and exerting anti-inflammatory effects [19]. In addition to whole-diet interventions, functional foods and synbiotic formulations, which combine fermentable fibers with live probiotic strains, are being explored for their capacity to boost SCFA output and modulate host immunity [21,49]. These strategies aim to restore a beneficial microbiota configuration conducive to mucosal homeostasis and tumor suppression.

Recent advances in engineered probiotics offer an innovative frontier in CRC prevention [48]. These designer strains are being developed to enhance site-specific butyrate delivery and achieve tumor-targeting capabilities, potentially offering a precise and sustainable means of microbiota-based chemoprevention [140].

## 6. Discussion

The gut microbiota and its metabolites, particularly SCFAs such as butyrate, have emerged as central regulators of CRC pathogenesis and prevention. Many studies have highlighted their roles in modulating epithelial integrity, immune surveillance, and epigenetic regulation within the colonic tumor microenvironment [12,15,92,125]. Among SCFAs, butyrate is consistently highlighted for its epigenetic activity, anti-inflammatory effects, and role in maintaining mucosal integrity [15,133,141]. However, despite encouraging mechanistic insights and clinical observations, several translational and methodological challenges must be addressed before microbiota-based interventions can be widely adopted in clinical practice.

### 6.1. SCFA and Probiotics in CRC

Butyrate, the most extensively studied SCFA in CRC, exerts anti-tumor effects through multiple mechanisms. These include HDAC inhibition, which alters gene transcription to suppress oncogenic pathways, and activation of G-protein coupled receptors (GPR41/43), which modulate mucosal immunity and apoptosis [87]. Additionally, SLC5A8-mediated transport enhances intracellular signaling in epithelial cells [133,134]. Preclinical studies with butyrate-producing bacteria such as *Butyricicoccus pullicaecorum* and *Clostridium butyricum* have demonstrated reduced tumor burden and enhanced mucosal integrity in CRC models [41,108].

Human studies, though promising, show heterogeneity due to differences in strain selection, dosage, and baseline microbiota. For instance, probiotics targeting *Faecalibacterium prausnitzii* and *Roseburia spp*. have shown potential in CRC prevention [120]. However, translational gaps remain, particularly regarding host-specific responses and regulatory approval pathways.

### 6.2. Critical Evaluation of Evidence

Preclinical models have robustly demonstrated the anticancer and immunoregulatory potential of probiotics and SCFAs. For example, *Butyricicoccus pullicaecorum* and *Clostridium* can reduce tumor burden, enhance mucosal integrity, and promote anti-tumor immunity [43,94,108]. However, these models often rely on high-dose exposures and simplified microbial communities that do not fully replicate the complexity of the human gut ecosystem. As a result, translating these promising findings into clinical outcomes remains challenging.

To clearly delineate the translational relevance of findings, we have synthesized key contrasts between animal and human studies in a new comparative summary table (Table 5). This table outlines differences in model systems, microbial interventions, SCFA modulation, mechanistic clarity, and observed outcomes across multiple domains, including tumor suppression, metabolic regulation, and neuropsychiatric effects [45,66,142]. By explicitly distinguishing preclinical from clinical data, this addition provides clarity on the extent to which animal-based evidence can be extrapolated to human populations.

Human clinical trials, though increasing in number, are frequently limited by heterogeneity in study design, including variations in probiotic strains, dosing regimens, intervention duration, and clinical endpoints. In cancer and neuropsychiatric disorders such as depression and anxiety, probiotics have shown preliminary benefits, yet the mechanistic basis for these effects remains insufficiently understood. Host-specific factors, such as genetics, diet, and baseline microbiota composition, likely contribute to inconsistent outcomes across studies [143,144].

In metabolic disorders, meta-analyses suggest that probiotics and SCFA-promoting interventions may modestly improve lipid profiles and glycemic control [47]. Nonetheless, strain-specific variability and individual host factors, including immune tone, microbiota diversity, and lifestyle, complicate the interpretation and generalization of results.

Importantly, even the same probiotic strain may elicit divergent physiological effects depending on the host’s baseline microbiome and environmental context [145,146]. These findings highlight the pressing need for stratified clinical trial designs, mechanistic studies, and personalized therapeutic approaches in microbiota-based research and intervention development. Key contrasts between animal models and human studies are summarized in Table 5, illustrating the translational gap and need for harmonized, mechanistically informed clinical research.

**Table 5 nutrients-17-02501-t005:** Representative preclinical and clinical studies on probiotic or SCFA-based interventions.

Study Context ^1^		Intervention	Key Outcomes ^2^	Limitations	Disease Focus/Reference
DSS-induced colitis (mouse)		Sodium butyrate	↑ Barrier integrity, ↓ TNF-α, ↑ Treg cells (via HDAC inhibition)	Acute inflammation model; lacks microbiota diversity	IBD/[94]
ApcMin/+ mouse model (CRC)		*Clostridium butyricum*	↓ Tumor burden, ↓ IL-6, ↑ Foxp3 expression, HDAC inhibition	No human microbiota or diet variability	CRC/[38]
CRC patients (perioperative RCT)		*L. rhamnosus GG*/ *B. longum*	↓ Postoperative infection, ↑ IL-10, improved epithelial barrier markers	Small cohort; mixed probiotic strains	CRC/[48]
Ulcerative colitis patients (RCT)		VSL#3 probiotic mixture	↑ Remission maintenance, ↓ IL-6 and TNF-α	Mixed response in Crohn’s vs. UC; strain-specificity unclear	IBD/[75]
IBS patients (RCT)		*L. plantarum*/*B. breve*	↓ Bloating and discomfort, improved quality of life	Placebo response possible; symptom-based endpoints	IBS/[115]
CKD mouse model	Synbiotic (inulin + *Bifidobacterium*)	↓ Uremic toxins, ↑ tight junction protein expression	Model does not replicate full human CKD complexity	CKD/[90]	
PD patients (pilot trial)		Multi-strain probiotic	↓ Constipation, ↑ SCFA levels, ↓ systemic inflammation	Short duration; no direct neurofunctional endpoints	PD/[107]
Neuropsychiatric disorder patients (RCT)		*L. helveticus*/*B. longum*	↓ Anxiety/depression scores; modulated cytokines (IL-6, TNF-α)	Small sample; microbiota composition not stratified	CNS/[147]

^1^ SCFA, short-chain fatty acid; CRC, colorectal cancer; IBD, inflammatory bowel disease; IBS, irritable bowel syndrome; PD, Parkinson’s disease; HDAC, histone deacetylase; CKD, chronic kidney disease; Treg, regulatory T cell; CNS, central nerve system; IL-6, interleukin 6; IL-10, interleukin 10; TNF-α, tumor necrosis factor-alpha; Foxp3, forkhead box p3; RCT, randomized controlled trial. 2 ↑, increased; ↓, decreased.

### 6.3. Methodological and Translational Limitations

#### 6.3.1. Methodological Limitations in the Literature

Despite the growing interest in microbiota-targeted therapies, the current body of literature is marked by substantial methodological limitations that impede the development of robust, translatable conclusions. One of the most pervasive issues is the reliance on small sample sizes in clinical trials, which compromises statistical power and limits the generalizability of findings [73,108]. Additionally, the short duration of follow-up in most studies restricts our understanding of the long-term efficacy and safety of microbiota-based interventions [140].

Another critical limitation lies in the lack of mechanistic insight. While many studies report improvements in clinical or biochemical outcomes, they often fail to elucidate the specific pathways through which individual probiotic strains or SCFAs exert their effects. This absence of mechanistic clarity hinders the identification of causative links and diminishes the translational impact of preclinical findings.

Heterogeneity in study design further complicates synthesis across the field. Outcome measures range widely, from microbial composition and SCFA levels to subjective symptom scores, making it difficult to compare results across studies or conduct meaningful meta-analyses. As a result, formulating standardized clinical recommendations remains challenging.

The promise of precision microbiome medicine, though compelling, remains largely theoretical [148]. Few trials incorporate personalized interventions based on host microbiota configurations or metabolomic profiles. Furthermore, there is little consensus regarding the standardization of probiotic formulations, dosages, or delivery methods, all of which affect reproducibility and therapeutic reliability.

Importantly, current regulatory frameworks present additional challenges. In many jurisdictions, probiotics are variably classified as dietary supplements, foods, or pharmaceutical agents, depending on their composition, intended use, and claims made. This regulatory ambiguity complicates product approval, quality control, labeling, and therapeutic standardization. As microbiota-based interventions move toward clinical application, harmonized global guidelines will be essential to streamline their evaluation, ensure safety, and support scalable innovation [30].

Together, in CRC, these methodological gaps are especially evident. While preclinical studies show that butyrate-producing microbes and SCFA-enhancing probiotics can inhibit tumor growth and support mucosal health, clinical translation has lagged. Most CRC trials are small, lack standardized probiotic protocols, and rarely assess mechanistic markers like SCFA levels or HDAC activity. Future studies should adopt CRC-specific designs integrating microbiome profiling and metabolomics to enable precision-guided interventions (Table 6).

#### 6.3.2. Inconsistencies and Failures in Probiotic Trials

Despite promising mechanistic studies and small-scale trials, clinical outcomes for probiotics have often been inconsistent across larger and more heterogeneous populations [149]. For example, randomized controlled trials (RCTs) in IBD, IBS, and CRC have reported mixed results [150,151,152]. While some studies show benefits in mucosal healing, symptom relief, or inflammatory marker reduction, others fail to demonstrate significant differences compared to placebo [116].

These inconsistencies stem from factors such as variation in probiotic strains and dosages, short study durations, lack of standardized endpoints, and host-specific differences in microbiota composition [149]. In many cases, the absence of mechanistic biomarkers (e.g., SCFA levels, cytokine profiles, colonization success) further limits interpretation.

As a result, future research must incorporate better patient stratification, strain-specific formulations, and multi-omic approaches to enhance reproducibility and translation. Standardizing study design, defining measurable mechanisms of action, and tracking patient-specific responses will be essential for clinical advancement.

### 6.4. Future Directions and Personalized Microbiome Therapeutics

An expanding array of microbiota-targeted strategies is shaping the future of precision medicine in gut health and systemic disease management. Among the most promising modalities are synbiotics, which combine prebiotics and probiotics to synergistically promote the growth and function of beneficial microbial taxa [46,80,90]. Postbiotics, composed of inactivated microbial cells and their bioactive metabolites, represent a safer and more stable alternative to live organisms while retaining key immunomodulatory and metabolic benefits.

Similarly, fecal microbiota transplantation (FMT) is gaining recognition, particularly for the treatment of refractory dysbiosis, such as in recurrent *Clostridioides difficile* infection [153]. FMT offers a holistic restoration of microbial diversity and function, although standardization and safety remain ongoing concerns.

Innovations in synthetic biology are enabling the design of engineered probiotics with programmable functions. These custom strains can be tailored to synthesize therapeutic molecules, modulate immune responses, or deliver drugs directly to mucosal tissues [21,94,108], paving the way for targeted microbiome-based therapeutics.

Despite these advances, several critical challenges must be addressed to ensure successful clinical translation.

#### 6.4.1. Rigorous Clinical Trials

Many current studies suffer from heterogeneity in microbial strains, dosing protocols, clinical endpoints, and patient populations. Well-designed, adequately powered randomized controlled trials with standardized methodologies and long-term follow-up are essential to validate safety, efficacy, and reproducibility [48,75,94,154]. In the context of CRC, engineered probiotics represent a promising frontier. Synthetic biology approaches have produced designer strains capable of targeted butyrate delivery or tumor microenvironment-specific modulation, opening new avenues for CRC prevention and therapy. These advances support a precision medicine framework for future microbiota-based CRC interventions.

#### 6.4.2. Integration of Multi-Omics

Incorporating metagenomics, transcriptomics, metabolomics, and immune profiling will be critical for understanding host–microbe interactions at a systems level [155,156]. These tools can facilitate the development of personalized microbiome therapies, stratifying patients based on features such as microbial enterotypes or SCFA output.

#### 6.4.3. Diet–Microbiota Interactions

Nutritional components such as dietary fibers, polyphenols, and omega-3 fatty acids substantially shape microbial composition and function [19,157,158,159]. Identifying synergistic combinations of dietary and microbial interventions may improve therapeutic efficacy and broaden clinical applications.

#### 6.4.4. Regulatory and Implementation Frameworks

To facilitate clinical adoption, the field must establish evidence-based guidelines, including quality control standards, dosing parameters, safety assessments, and clearly defined therapeutic endpoints. One of the most pressing barriers to clinical implementation is regulatory ambiguity surrounding the classification of probiotics (e.g., supplement vs. drug) continues to be a barrier to widespread deployment [74]. In various countries, probiotics may be categorized as dietary supplements, functional foods, or pharmaceutical agents depending on the jurisdiction, intended health claims, and formulation characteristics. This inconsistency leads to fragmented oversight, varied safety requirements, and inconsistent standards for efficacy evaluation.

Such classification disparities not only delay the translation of promising microbiota-based therapies into clinical settings but also complicate large-scale manufacturing, labeling, and clinician–patient communication. For example, probiotics marketed as supplements may not require the rigorous clinical trial validation expected of pharmaceutical agents, which can undermine clinician confidence and hinder insurance reimbursement policies. Conversely, framing probiotics strictly as drugs may impose regulatory burdens that stifle innovation.

Therefore, the establishment of harmonized international frameworks that delineate probiotic categories, define functional claims, and align safety and efficacy assessment criteria is critical. Clear regulatory pathways will enhance product quality assurance, facilitate therapeutic development, and support scalable integration of microbiome-based interventions into mainstream medicine [160,161].

## 7. Conclusions

The gut microbiota, its key metabolites such as SCFAs (especially butyrate), and probiotic strategies are integral to human health and disease, particularly in the context of CRC, as highlighted throughout this review [21,90,100,141]. These elements contribute collectively to the maintenance of intestinal homeostasis, the regulation of systemic inflammation, and the modulation of host immunity and metabolism.

Among SCFAs, butyrate stands out for its pleiotropic actions, which include reinforcing epithelial barrier integrity, suppressing pro-inflammatory cytokines, modulating gene expression via epigenetic pathways, and exerting anti-tumor effects, especially in the context of CRC [38,141]. Concurrently, probiotics have emerged as a promising intervention for restoring microbial balance, enhancing mucosal repair, and stimulating endogenous SCFA production. These benefits extend to a range of conditions spanning gastrointestinal, metabolic, neuropsychiatric, renal, and oncological domains [162].

However, the clinical translation of microbiota-targeted strategies remains challenged by response heterogeneity. Factors such as host genetics, dietary background, baseline microbial diversity, and immune tone significantly influence therapeutic outcomes [30,60,114,140]. Moreover, many current trials are constrained by small sample sizes, inconsistent strain selection, variable dosages, and limited follow-up duration [31,73,102,154].

Future directions must prioritize the advancement of precision microbiome medicine, a model integrating microbiome sequencing, dietary modulation, immunometabolic profiling, and omics-based diagnostics. Emerging therapeutic modalities such as synbiotics, postbiotics, engineered probiotics, and metabolite-targeted approaches hold substantial promise but require rigorous validation in large-scale, standardized clinical trials [30,31].

Importantly, microbiota-targeted interventions should be viewed as complementary rather than replacement therapies, and ideally embedded within broader, personalized treatment frameworks that also incorporate nutritional, pharmacological, and diagnostic strategies [144]. Realizing the full therapeutic potential of the microbiome will depend on mechanistic clarity, therapeutic standardization, and regulatory alignment, as well as continued investment in interdisciplinary translational research.

In conclusion, the gut microbiome represents a compelling frontier for the next generation of personalized healthcare. Unlocking its full clinical potential will require mechanistic clarity, therapeutic standardization, and integration with dietary and omics-based strategies. A concise synthesis of these insights is illustrated in Figure 2, which highlights the central roles of the gut microbiota, SCFAs (notably butyrate), and probiotics, along with key future directions and implementation challenges.

## Figures and Tables

**Figure 1 nutrients-17-02501-f001:**
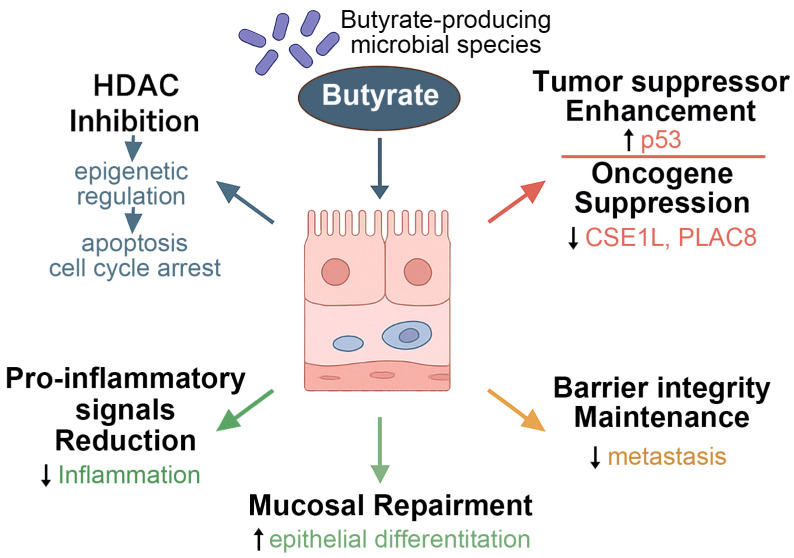
Butyrate-mediated regulation of epithelial differentiation and tumor suppression, showing the multifaceted roles of butyrate-producing microbial species in modulating host cellular functions. Butyrate inhibits HDACs, leading to epigenetic regulation that suppresses oncogenes (e.g., CSE1L, PLAC8) and enhances tumor suppressor gene expression. These actions contribute to cell cycle arrest, promotion of apoptosis, and inhibition of metastasis. Additionally, butyrate supports epithelial differentiation, reinforces mucosal barrier integrity, and reduces pro-inflammatory signaling, thereby maintaining tissue homeostasis and facilitating mucosal repair. HDACs, histone deacetylases; CSE1L, chromosome segregation 1 like; PLAC8, placenta-associated 8.

**Figure 2 nutrients-17-02501-f002:**
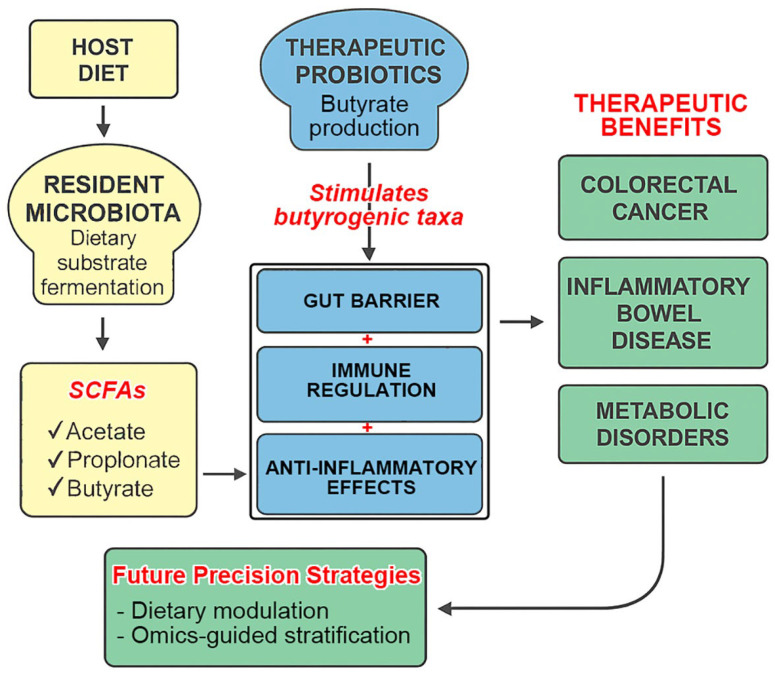
Integrated model of microbiota–SCFA–probiotic interactions and precision therapeutic strategies. This diagram synthesizes the interconnected roles of resident microbiota, therapeutic probiotics, and short-chain fatty acids (SCFAs) in maintaining gut health and modulating systemic immunity and inflammation. Microbial fermentation of dietary substrates produces SCFAs, particularly butyrate, which mediate gut barrier integrity, immune regulation, and anti-inflammatory effects. Therapeutic probiotics enhance butyrogenic taxa and support endogenous SCFA production. These mechanisms contribute to therapeutic benefits in conditions such as colorectal cancer, inflammatory bowel disease, and metabolic disorders. Future precision strategies, including dietary modulation, omics-guided stratification, and regulatory alignment, are essential to unlock the full translational potential of microbiome-based therapies.

**Table 2 nutrients-17-02501-t002:** Summary of probiotic effects in human diseases.

Disease	Effects of Probiotics ^1^	Reference
Gastrointestinal disorders	Reduces inflammation, improves bowel habits, shortens diarrhea duration	[46]
Metabolic disorders	Lowers LDL cholesterol, body weight, waist circumference	[47]
Colorectal cancer	Restores microbial balance, enhances immune response	[48]
Neuropsychiatric conditions	Alleviates depression, anxiety, modulates neurotransmitters	[42]
General immunity	Boosts SCFA production, reduces systemic inflammation	[49]

^1^ LDL, low-density lipoprotein; SCFA, short-chain fatty acid.

**Table 3 nutrients-17-02501-t003:** Integrated summary of probiotic and SCFA effects in human diseases ^1^.

Disease ^2^	Probiotic Effect ^3^	SCFA Role ^4^	Outcome ^5^	Reference
CKD	Restores gut integrity; reduces uremic toxins	Enhances barrier function; reduces inflammation via GPR109A	Improved renal and GI function	[90]
PD	Modulates gut–brain axis; alters microbial composition	Regulates neuroinflammation via SCFA-GPR signaling	Reduced neurodegeneration	[21]
IBD	Reduces inflammation, strengthens mucosal barrier	Butyrate promotes epithelial healing and Treg differentiation via HDAC inhibition	Disease symptom alleviation	[94]
CVDs	Modulates lipid metabolism; lowers systemic inflammation	Supports vascular homeostasis and lipid regulation; reduces LPS translocation	Reduced cardiometabolic risk	[80]
MetS	Lowers weight and LDL; modulates microbiota	Regulates metabolism through PPAR-γ and GPR43 activation	Improved metabolic profile	[47]
CRC	Enhances immune function; restores microbial balance	Butyrate induces apoptosis and Foxp3 expression via HDAC inhibition; suppresses oncogenes (e.g., CSE1L, PLAC8)	Tumor suppression and barrier protection	[48]

^1^ SCFA, short-chain fatty acid. ^2^ CKD, chronic kidney disease; PD, Parkinson’s disease; IBD, inflammatory bowel disease; CVDs, cardiovascular diseases; MetS, metabolic syndrome; CRC, colorectal cancer. ^3^ LDL, low-density lipoprotein. ^4^ GPR109A, G-protein coupled receptor 109A; GPR43, G-protein coupled receptor 43; PPAR-γ, peroxisome proliferator activated receptor gamma; Foxp3, Forkhead box p3; HDAC, histone deacetylase; CSE1L, chromosome segregation 1 like; PLAC8, placenta associated 8. ^5^ GI, gastrointestinal.

**Table 4 nutrients-17-02501-t004:** Butyrate-producing bacteria and their effects on colorectal cancer.

Gut Microbes	Mechanism ^1^	Health Effects ^2^	Reference
*Butyricicoccus pullicaecorum*	Upregulates SLC5A8, GPR43 Downregulates CSE1L expression	Reduces CRC progression Reverses p53-related genetic distortion	[43] [38]
*Clostridium butyricum*	Enhances mucosal immunity	Alleviates inflammation, supports colon health	[108]
*Roseburia* spp.	Butyrate production affected by antibiotics	Influences intestinal barrier and CRC risk	[18]

^1^ SLC5A8, solute carrier family 5 member 8; GPR43, G protein-coupled receptor 43; CSE1L, chromosome segregation 1 Like. ^2^ CRC, colorectal cancer.

**Table 6 nutrients-17-02501-t006:** Key methodological limitations in microbiota-based clinical research.

Domain	Limitation ^1^	Implication
Sample size	Predominantly small cohorts in clinical trials	Reduces statistical power and limits the generalizability of findings
Follow-up duration	Short intervention and monitoring periods	Hinders evaluation of long-term efficacy, durability, and safety
Mechanistic insight	Lack of mechanistic endpoints (e.g., SCFA quantification, gene expression)	Obstructs pathway-level understanding and weakens translational value
Outcome heterogeneity	Inconsistent endpoints across studies (e.g., microbiota composition vs. symptoms)	Limits cross-study comparability and reduces feasibility of reviews or meta-analyses
Personalization	Limited use of host microbiome or metabolomic stratification	Impedes development of targeted or precision interventions
Intervention standardization	Variability in probiotic strains, dosages, formulations, and delivery systems	Undermines reproducibility and clinical reliability
Regulatory ambiguity	Unclear classification of probiotics (e.g., supplement vs. drug); lack of harmonized international standards	Delays clinical translation, complicates product approval, scalability, and quality control; creates uncertainty for clinicians and policymakers.

^1^ SCFA, short-chain fatty acid.

## Data Availability

The original contributions presented in this study are included in the article. Further inquiries can be directed to the corresponding authors.

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
