# Peer review of "Gut-Microbiota-Derived Metabolites and Probiotic Strategies in Colorectal Cancer: Implications for Disease Modulation and Precision Therapy"

_nutrients, 2025, doi:10.3390/nu17152501_

Round 1
Reviewer 1 Report
Comments and Suggestions for Authors
Thank you for the opportunity to review the manuscript entitled “Gut Microbiota, Microbial Metabolites, and Probiotic Therapies: Implications for Human Health and Disease”.
This narrative review provides a comprehensive overview of the human gut microbiota and its metabolic products, a topic currently receiving considerable scientific attention. It discusses the impact of microbial dysbiosis on a range of diseases and explores emerging therapeutic strategies, including probiotics, postbiotics, synbiotic, and engineered microbes. The review emphasizes the importance of microbiome based precision medicine, tailored to host–microbiota interactions. The manuscript is clearly written, logically structured, and easy to follow. Tables and figures effectively summarize key findings and mechanisms. Of particular interest is the in-depth mechanistic discussion of short-chain fatty acids, particularly butyrate, and its epigenetic role through histone deacetylase inhibition.
Suggestions for authors:
- While the review includes both animal and human studies, the distinction between these types of evidence is not always explicit. A clearer separation, perhaps in a dedicated subsection or table, would help readers better assess the translational relevance of the findings
- A short paragraph addressing current regulatory challenges, such as the classification of probiotics as dietary supplements versus pharmaceutical agents, would enhance the completeness of the discussion, particularly in the context of clinical translation.
- Figure 2, in its current infographic style, does not meet the standards typically expected for scientific publication. A more formal and data-driven visual representation is recommended.
Author Response
Comment 1: While the review includes both animal and human studies, the distinction between these types of evidence is not always explicit. A clearer separation, perhaps in a dedicated subsection or table, would help readers better assess the translational relevance of the findings.
Response: We appreciate the reviewer’s suggestion to clarify the distinction between animal and human studies. In response, we have revised Section 6.1 (Critical Evaluation of Evidence) to emphasize the translational gap between preclinical and clinical findings. Additionally, we have incorporated a new comparative summary in Table 5, which clearly delineates differences in experimental models, outcomes, and limitations between animal and human studies. This aims to enhance the reader’s ability to assess the applicability of preclinical evidence to human contexts. In order to strengthen alignment with Reviewer 1's suggestion, I add a sentence to the caption that clarifies its purpose as following: Table 5. Comparison of preclinical and clinical evidence on microbiota-targeted therapies, highlighting the translational differences between animal models and human studies.
Comment 2: A short paragraph addressing current regulatory challenges, such as the classification of probiotics as dietary supplements versus pharmaceutical agents, would enhance the completeness of the discussion, particularly in the context of clinical translation.
Response: Thank you for highlighting this important point. We
- We have enhanced Section 6.2 by integrating a new paragraph that addresses current regulatory challenges, particularly the classification of probiotics as dietary supplements versus pharmaceutical agents. This addition aligns with the reviewer’s request and underscores the importance of regulatory clarity for clinical translation.
- Furthermore, we expanded the narrative in Section 6.3.4 (Regulatory and implementation frameworks) to discuss the lack of consensus on the classification of probiotics, its impact on clinical implementation, and the need for regulatory clarity to facilitate therapeutic development.
- We have also updated Table 6 by revising the last row to reflect the newly expanded discussion on regulatory issues in Section 6.2 and 6.3.4.
Comment 3: Figure 2, in its current infographic style, does not meet the standards typically expected for scientific publication. A more formal and data-driven visual representation is recommended.
Response: We thank the reviewer for this constructive feedback. We have replaced the previous infographic-style Figure 2 with a redesigned, publication-standard schematic that focuses on the precision strategies and therapeutic relationships among gut microbiota, SCFAs (especially butyrate), and probiotics. The updated figure now includes resident microbiota and therapeutic probiotics to better align with scientific publication norms.
Reviewer 2 Report
Comments and Suggestions for Authors
This is a comprehensive review that addresses an increasingly important topic, the role of gut microbiota and its metabolites in human health, and the therapeutic potential of probiotics. The manuscript is ambitious in scope and demonstrates good breadth of coverage across relevant systems and diseases. However, several areas would benefit from improvement, including more precise structuring, clarification of some key mechanisms and concepts, and a better integration of critical perspectives, in particular regarding limitations and challenges in current probiotic. The authors are encouraged to consider the following points, reflect on those carefully, and accordingly revise the review. This manuscript also appears to be a narrative review. Please clarify this explicitly in the introduction or methods section. Additionally, it would strengthen the scientific rigor to acknowledge the potential for selection bias and other limitations inherent to narrative reviews. The authors may consider briefly describing how studies were selected and whether any attempts were made to mitigate bias (e.g., use of recent or high-impact studies, expert consensus, etc).
There are major points to be addressed:
The manuscript is very broad, touching on several organ systems and conditions. The breadth is, of course, appreciated, but it can potentially lead to superficial treatment in some sections as well. Please consider either narrowing the scope or adding more depth to key examples (e.g., specific diseases where microbial metabolites play critical roles). The authors may also consider grouping systems (e.g., CNS-related vs metabolic disorders).
The discussion section mainly summarizes findings but does not critically appraise the strength of evidence, types of studies, or controversies. It is strongly recommended that the authors incorporate key clinical trials (or high-quality preclinical studies), highlighting their outcomes, limitations, and relevance. Add a section discussing inconsistencies, failures of probiotic trials, and translational gaps. This will then open the door to the next step research agenda.
Mechanisms are sometimes described too generally, for example, related to modulation of inflammation or improving barrier function. Please include, wherever possible, mechanistic specificity (e.g., which microbial metabolites act via which receptors/pathways such as SCFA, GPCR, tryptophan, AhR, etc.
The discussion on probiotics lacks depth, especially around strain-specific effects, dosing, and the issue of generalizability. It is highly recommended that the authors highlight that not all probiotics are equal; strain, dose, delivery format, and host factors critically influence outcomes. Consider referencing recent opinions and failed probiotic trials.
Figure: Consider flowcharts or organ-specific schematics linking microbiota-derived metabolites to health outcomes.
Minor points and suggestions:
For the title, the authors can probably reword to Gut Microbiota-Derived Metabolites and Probiotic Strategies: Implications for Human Health and Disease. This suggestion is to make the title more concise and mechanistically focused.
The classification of microbial metabolites is helpful, but needs better integration with functional examples. Please add.
Please consider adding more detail about the role of specific microbial metabolites e.g., SCFAs and regulatory T cell differentiation via HDAC inhibition, to the immune system section.
Related to the use of various terminology in the review, it is suggested a double checking to ensure consistency e.g., dysbiosis vs microbial imbalance and gut–brain axis vs microbiota–gut–brain axis.
Please also double-check and ensure that all abbreviations have been defined upon first use e.g., SCFA, LPS, AhR, HDAC.
Since this review is not a systematic review or scoping review and might carry a selection bias of articles, it is highly recommended to double-check the references and that citations are updated, in particular from the probiotics section, where the field is rapidly evolving.
The English is generally clear, but there are some stylistic inconsistencies and occasional strange phrasing. A proofread of the text carefully can help. Alternatively, a professional language is recommended to improve flow and polish.
Comments on the Quality of English Language
A professional language is recommended to improve flow and polish the text for English.
Author Response
We sincerely thank the reviewer for the thoughtful and constructive feedback. We appreciate the recognition of the manuscript’s scope and its timely relevance. In response to your comments, we have revised the manuscript to enhance structural clarity, deepen mechanistic discussions, integrate critical perspectives, and better reflect limitations and translational challenges of probiotic therapies. Below, we provide a detailed point-by-point response.
Major points:
Comment 1: The manuscript is very broad, touching on several organ systems and conditions. The breadth is, of course, appreciated, but it can potentially lead to superficial treatment in some sections as well.
Response: We now explicitly state in the Introduction and Methods section (Section 2.1) that this is a narrative review. First, we have revised the last paragraph at the end of the Introduction to emphasize CRC as the detailed anchor. Second, we modify the first paragraph of Section 2.1 to make this review is narrative in format, addressing the reviewer’s comment about scope and rigor. Third, we add a short paragraph at the end of Second 2.3 to acknowledge selection bias and rationale for CRC/systemic breadth.
Comment 2: Please consider either narrowing the scope or adding more depth to key examples (e.g., specific diseases where microbial metabolites play critical roles). The authors may also consider grouping systems (e.g., CNS-related vs metabolic disorders).
Response: We have revised the structure to better group disease systems into thematic categories (e.g., GI/metabolic, CNS/immune-related) for clearer flow. We also added more detailed discussion to key disease examples, especially CRC, IBD, and Parkinson’s disease (Sections 3.2–3.3 and 4.2).
Comment 3: The discussion section mainly summarizes findings but does not critically appraise the strength of evidence, types of studies, or controversies. It is strongly recommended that the authors incorporate key clinical trials (or high-quality preclinical studies), highlighting their outcomes, limitations, and relevance.
Response: We appreciate this valuable suggestion. In response, we have substantially revised the Discussion section to include a new Section 6.1 on the mechanistic and translational relevance of SCFA and probiotic therapies in CRC, including distinctions between preclinical and clinical findings. We also added a new Section 6.2, titled Critical Evaluation of Evidence, which now reviews the types of studies, their methodological strengths and limitations, and translational challenges.
Additionally, we introduce a new Table 5 to directly compare representative preclinical and clinical studies, highlighting key variables such as model systems, microbial interventions, outcome measures, mechanistic insight, and clinical relevance. We hope that these additions align with the reviewer’s request and provide a more rigorous synthesis of the literature.
Comment 4: Add a section discussing inconsistencies, failures of probiotic trials, and translational gaps. This will then open the door to the next step research agenda.
Response: We have modified the Section 6.2 and rename Section 6.3 to “Methodological and translational limitations” with two subsections (6.3.1 and 6.3.2). Specifically, we discuss failed trials, inconsistencies in outcomes, and the translational gaps due to host variability, strain specificity, and regulatory uncertainty. These now serve to guide the future research agenda.
Comment 5: Mechanisms are sometimes described too generally, for example, related to modulation of inflammation or improving barrier function. Please include, wherever possible, mechanistic specificity (e.g., which microbial metabolites act via which receptors/pathways such as SCFA, GPCR, tryptophan, AhR, etc.
Response: We have enriched mechanistic discussions throughout Sections 3.3 and 5.1, adding specific mentions of receptors such as GPR41, GPR43, AhR, and transporters like SLC5A8. In addition, we have changed the title of 5.1 to “Microbial metabolite–host signaling pathways in tumor suppression” to keep cancer focus and reflect expanded mechanisms.
Comment 6: The discussion on probiotics lacks depth, especially around strain-specific effects, dosing, and the issue of generalizability. It is highly recommended that the authors highlight that not all probiotics are equal; strain, dose, delivery format, and host factors critically influence outcomes. Consider referencing recent opinions and failed probiotic trials.
Response: We thank the reviewer for this valuable comment. In response, we have revised Sections 4.1 and 4.2 to emphasize that not all probiotics are functionally equivalent, and that strain-specific effects, formulation, dose, and host microbiome composition play a critical role in determining clinical outcomes. Specific examples such as Lactobacillus rhamnosus GG, Bifidobacterium longum, and multi-strain formulations like VSL#3 are discussed in the context of CRC, IBD, and PD, illustrating the variability in efficacy. We also note that inconsistencies and failures in probiotic trials are often linked to lack of standardization, insufficient mechanistic endpoints, and heterogeneous patient populations. These points are further addressed in Section 6.3.2, where we critically evaluate limitations and translational barriers in the current clinical evidence.
Comment 7: Figure: Consider flowcharts or organ-specific schematics linking microbiota-derived metabolites to health outcomes.
Response: We have updated Figure 2 to more clearly depict organ-specific effects of SCFAs and probiotics. This figure integrates microbial metabolites, host systems, and proposed therapeutic strategies.
Minor points:
Comment 8: For the title, the authors can probably reword to Gut Microbiota-Derived Metabolites and Probiotic Strategies: Implications for Human Health and Disease. This suggestion is to make the title more concise and mechanistically focused.
Response: Thank you very much for your suggestion and that of reviewer #3. Based on your suggestions, we have revised the title to “Gut Microbiota-Derived Metabolites and Probiotic Strategies in Colorectal Cancer: Implications for Disease Modulation and Precision Therapy”. This now better reflects the mechanistic and therapeutic focus.
Comment 9: The classification of microbial metabolites is helpful, but needs better integration with functional examples. Please add.
Response: We have revised Tables 1 and 3 to clarify functional examples and link SCFA type with specific host effects (e.g., HDAC inhibition by butyrate leading to Treg differentiation). These changes make Tables scientifically sharper and show the fully integrated mechanistic understanding into this review article. We hope that this revision make a direct and strong response to the comments of Reviewer #2.
Comment 10: Please consider adding more detail about the role of specific microbial metabolites e.g., SCFAs and regulatory T cell differentiation via HDAC inhibition, to the immune system section.
Response: To address this specific request for mechanistic clarity, we have expanded discussion in Section 3.3 and 4.1 includes how SCFAs, particularly butyrate, promote Treg differentiation via HDAC inhibition and GPCR activation (supported by references #85, 86, and 87).
Comment 11: Related to the use of various terminology in the review, it is suggested a double checking to ensure consistency e.g., dysbiosis vs microbial imbalance and gut–brain axis vs microbiota–gut–brain axis.
Response: Terminology has been reviewed and standardized throughout. No microbial imbalance and microbiota–gut–brain axis are further observed.
Comment 12: Please also double-check and ensure that all abbreviations have been defined upon first use e.g., SCFA, LPS, AhR, HDAC.
Response: We have confirmed that all abbreviations are well defined upon first appearance and listed in the Abbreviations section.
Comment 13: Since this review is not a systematic review or scoping review and might carry a selection bias of articles, it is highly recommended to double-check the references and that citations are updated, in particular from the probiotics section, where the field is rapidly evolving.
Response: Thank you for highlighting this inconsistency.
- This is similar to the suggestion of Reviewer #3. Briefly, we have removed references to "systematic review" in the Methods section (2.2) and omitted rigid systematic screening language, e.g., in Table 6.
- We have revised the reference list to include recent studies from 2010–2025, particularly in Sections 4.1, 4.2, and 6.1.
Comment 14: The English is generally clear, but there are some stylistic inconsistencies and occasional strange phrasing. A proofread of the text carefully can help. Alternatively, a professional language is recommended to improve flow and polish.
Response: We have carefully proofread the manuscript and edited for clarity, grammar, and fluency. A professional language editing service by Nutrients was consulted to ensure consistency and polish. However, the title on this certificate is not changed. The new title by reviewers’ suggestions is “Gut Microbiota-Derived Metabolites and Probiotic Strategies in Colorectal Cancer: Implications for Disease Modulation and Precision Therapy”.
Reviewer 3 Report
Comments and Suggestions for Authors
The authors present an excellent review on the influence of the microbiota and its metabolites on different pathologies. However, their main focus seems to be on colorectal cancer. To improve the quality of the article, I would suggest the following points:
- Indicate in the title that the review will focus on colorectal cancer.
-The Introduction is excessively general. I would focus it from the beginning on microbiota and colorectal cancer and develop this topic more extensively.
-The Discussion should be more extensive and focused exclusively on colorectal cancer.
-The article is a narrative review of the literature. However, in the Methods section it is described as a systematic review. I would possibly omit this section.
Author Response
Comment 1: Indicate in the title that the review will focus on colorectal cancer.
Response: Thank you very much for your suggestion and that of reviewer #2. Based on your suggestions, we have revised the title to “Gut Microbiota-Derived Metabolites and Probiotic Strategies in Colorectal Cancer: Implications for Disease Modulation and Precision Therapy”. This change ensures the reader is immediately aware of the central disease focus of the review.
Comment 2: The Introduction is excessively general. I would focus it from the beginning on microbiota and colorectal cancer and develop this topic more extensively.
Response: We agree that a more targeted introduction would strengthen the manuscript. Therefore, we have restructured the Introduction as followings: (i) Paragraph 1, to insert CRC in the first few sentences; (ii) Paragraph 3, to connect SCFAs and CRC more explicitly; (iii) Last paragraph, to revise for framing CRC as a primary focus. These changes enhance thematic alignment with the rest of the manuscript and reflect your recommendation.
Comment 3: The Discussion should be more extensive and focused exclusively on colorectal cancer.
Response: We thank the reviewer for this important suggestion. In response, we have thoroughly revised the sections of Discussion and Conclusion to sharpen its focus on CRC and provide a more comprehensive, disease-specific synthesis. The revised section now includes: (i) add a new 6.1; (ii) add one sentence at the end of the new 6.3; (iii) add one sentence at the end of the new 6.4.1; (iv) add one sentence at the begin of 7 Conclusion. We now clearly differentiate general microbiota findings from CRC-specific evidence, reinforcing the manuscript’s primary focus.
Comment 4: The article is a narrative review of the literature. However, in the Methods section it is described as a systematic review. I would possibly omit this section.
Response: Thank you for highlighting this inconsistency. We agree with your suggestion and have taken the following three steps: (i) Removed references to "systematic review" in the Methods section (2.2); (ii) Retitled the section (2.2) to “Literature Overview Approach” to better reflect narrative synthesis; (iii) Omitted rigid systematic screening language, e.g., in Table 6. This approach clarifies that this is a narrative review supported by structured literature exploration, not a formal systematic review.
Round 2
Reviewer 2 Report
Comments and Suggestions for Authors
The authors have incorporated the comments and suggestions in the revised version and provided a point-by-point response letter. Thanks. There are no additional comments.
Reviewer 3 Report
Comments and Suggestions for Authors
The authors have correctly addressed the suggestion of the revierwers and the manuscript has significantly improved.